# Vulnerability of agriculture to climate change increases the risk of child malnutrition: Evidence from a large-scale observational study in India

Bidhubhusan Mahapatra[1]*, Monika Walia[2], Chitiprolu Anantha Rama Rao[3]ᵒ‡, Bellapukonda Murali Krishna Raju[3]ᵒ, Niranjan Saggurti[1]ᵒ‡

1 Population Council, Zone 5A, India Habitat Centre, New Delhi, India, 2 International Food Policy Research Institute, New Delhi & Ex-Population Council, New Delhi, India, 3 ICAR-Central Research Institute for Dryland Agriculture, Santoshnagar, Saidabad, Hyderabad, India

ᵒ These authors contributed equally to this work.
‡ CARR and NS are joint senior authors on this work.
* bbmahapatra@gmail.com, bbmahapatra@popcouncil.org

**Data Availability Statement:** The NFHS-4 data is available at the DHS website and can be downloaded from https://dhsprogram.com/data/

## Abstract

### Introduction

The impact of climate change on agriculture and food security has been examined quite thoroughly by researchers globally as well as in India. While existing studies provide evidence on how climate variability affects the food security and nutrition, research examining the extent of effect vulnerability of agriculture to climate change can have on nutrition in India are scarce. This study examined a) the association between the degree of vulnerability in agriculture to climate change and child nutrition at the micro-level b) spatial effect of climate vulnerability on child nutrition, and c) the geographical hotspots of both vulnerability in agriculture to climate change and child malnutrition.

### Methods

The study used an index on vulnerability of agriculture to climate change and linked it to child malnutrition indicators (stunting, wasting, underweight and anaemia) from the National Family Health Survey 4 (2015–16). Mixed-effect and spatial autoregressive models were fitted to assess the direction and strength of the relationship between vulnerability and child malnutrition at macro and micro level. Spatial analyses examined the within-district and across-district spill-over effects of climate change vulnerability on child malnutrition.

### Results

Both mixed-effect and spatial autoregressive models found that the degree of vulnerability was positively associated with malnutrition among children. Children residing in districts with a very high degree of vulnerability were more like to have malnutrition than those residing in districts with very low vulnerability. The analyses found that the odds of a child suffering from stunting increased by 32%, wasting by 42%, underweight by 45%, and anaemia by

available-datasets.cfm. The climate data can be accessed by giving request at: http://dsp.imdpune. gov.in/. The authors had no special access privileges to the data others would not have.

**Funding:** The author(s) received no specific funding for this work.

**Competing interests:** The authors have declared that no competing interests exist.

63% if the child belonged to a district categorised as very highly vulnerable when compared to those categorised as very low. The spatial analysis also suggested a high level of clustering in the spatial distribution of vulnerability and malnutrition. Hotspots of child malnutrition and degree of vulnerability were mostly found to be clustered around western-central part of India.

## Conclusion

Study highlights the consequences that vulnerability of agriculture to climate change can have on child nutrition. Strategies should be developed to mitigate the effect of climate change on areas where there is a clustering of vulnerability and child malnutrition.

## Introduction

Climate change is probably the most complex and challenging environmental problem faced by the world today and is increasingly being recognized as a potent threat to agriculture in general, and specifically to food security [1, 2]. Climate scientists have predicted that climate change is going to have a significant impact on agriculture which will ultimately affect the quality and quantity of food production [1, 3]. It is estimated that agricultural output in developing countries will decline by 10–20% by 2080 [4]. This will have adverse consequences in achieving universal food security and meeting the nutritional requirement of communities [2, 5, 6]. Estimates suggest that with the changing climate, in 2050, there will be 62% more severe stunting cases than what could be without any change in the current climatic scenario [7]. Currently, about one billion people are deprived of enough food [8], over 150 million children are stunted, and another 50 million are wasted [9]. Though recent evidence suggests that there have been some improvements in nutritional indicators, climate change can undermine ongoing efforts to reduce hunger and enhance food security [7, 10]. The situation in India is much like the global scenario where with changing climate and ever-growing population, the demand for food is bound to increase further. An increase in 1–2˚C in temperature is going to have a negative impact on the yield of major cereal crops in low altitude countries like India [11] which in turn will impact the nutritional status of the population [12].

The literature review for this study focused on reviewing documents on issues of climate change, agriculture, food security, and nutrition. The literature search suggests that there have been several studies globally and in the Indian context that have examined the impact of climate change on agriculture and food security. The available body of evidence estimating impact of climate change on agriculture, food security and nutrition have documented the impact of rainfall and temperature variability (including level and pattern) as well as of extreme weather events on undernutrition among children [3, 13–17]. A study conducted in Mali, Africa found that by 2025, due to climate and livelihood changes an additional million children will be exposed to increased risk of malnutrition [13]. Similarly, a longitudinal study conducted in Ethiopia between 1996–2004 estimated that while one standard deviation (SD) increase in rainfall may lead to 0.24 SD increase in moderate stunting, one SD increase in temperature may lead to 0.22 SD decrease in moderate stunting [13, 14]. In Indian context, studies examining the impact of climate change on malnutrition found that children in flood affected households were twice more likely to be stunted and underweight compared to their counterparts living in non-flooded areas. Research specific to India suggests that with the current level of crop yields remaining constant till 2050, there will be a severe shortage of micronutrient

supply to the households [18]. Prior research has also examined the impact of rainfall and temperature variability (including level and pattern) on undernutrition among children. While existing studies provide evidence on how climate variability affects the food security and nutrition, there has been dearth of research examining the extent of effect vulnerability of agriculture to climate change can have on nutrition in India. The current study contributes to existing body of evidence on climate change and nutrition by assessing whether vulnerability of agriculture to climate change is linked to the nutritional status of communities. The study aims to answer three research questions: (i) Is there an association between the degree of vulnerability in agriculture to climate change and child nutrition at the micro-level? (ii) Is there any spatial effect of climate vulnerability on child nutrition? and (iii) Which are the geographical hotspots of both vulnerability in agriculture to climate change and child malnutrition?

## Methods

### Data

The study used two data sources: (i) climate vulnerability index developed under National Initiative on Climate Resilient Agriculture (NICRA) project of the Indian Council of Agricultural Research (ICAR) [12, 19] and (ii) children's nutritional status derived from National Family Health Survey 4 (NFHS-4).

**Climate vulnerability index.**   Intending to make Indian agriculture resilient to climate change, ICAR launched the NICRA in the year 2011. Per Intergovernmental Panel on Climate Change (IPCC), the NICRA study considered the integrated approach to define vulnerability as "a function of the extent and degree to which an entity is exposed, the sensitivity of the entity to climate change and adaptive capacity to adapt to and cope with the changing climate" [20]. The study used 38 indicators from various sources to construct the index on vulnerability of agriculture to climate change for 572 Indian districts (as per the Census of India 2001). These indicators were chosen and assigned to the three components of vulnerability–sensitivity, exposure, and adaptive capacity based on literature review, their relevance, and discussions with subject experts (Table 1). Selection of these 38 indicators was based on the extent and intensity of the effect of climate change and/or variability as reflecting its sensitivity. For example, indicators such as net sown area and rural population density (Table 1) determine the extent of the problem whereas the indicators such as water holding capacity of soil, frequency and intensity of occurrence of climate shocks determine the intensity or degree of effect of such shock. Likewise, indicators that are relatively more responsive to policy measures were considered for the adaptive capacity component of vulnerability. Indicators under each component of vulnerability were first normalized using the following min-max formula. When the indicator was positively related to the index, the formula used was:

$$Z_i = \frac{X_i - X_{min}}{X_{max} - X_{min}}$$

When the indicator was negatively related to the index, the formula used was:

$$Z_i = \frac{X_{max} - X_i}{X_{max} - X_{min}}$$

where $Z_i$ = normalized value of $i^{th}$ district with respect to the indicator X
$X_i$ = value of indicator in original units for $i^{th}$ district
$X_{min}$ = minimum value of the indicator in original units across the districts
$X_{max}$ = maximum value of the indicator in original units across the districts

**Table 1. Indicators of sensitivity, exposure and adaptive capacity used for computing vulnerability index.**

| Sensitivity | Exposure | Adaptive capacity |
|---|---|---|
| Net sown area in relation to geographical area (%) | Change (%) in annual rainfall during mid-century (2021–50) relative to the baseline (1961–90) | Rural poor defined as the % of rural population that is below poverty line |
| Extent of degraded and waste lands in relation to geographical area (%) | Change (%) in June rainfall during mid-century (2021–50) relative to the baseline (1961–90) | SC/ST population (%) |
| Average annual rainfall (mm) | Change (%) in July rainfall during mid-century (2021–50) relative to the baseline (1961–90) | Workforce in agriculture defined as % of workers engaged in agriculture in relation to total workers |
| Cyclone proneness constructed by combining the number of cyclones crossing the district, number of severe cyclones crossing the district, probable maximum precipitation for a day, probable maximum winds in knot, probable maximum storm surge | Change (%) in number of rainy days during mid-century (2021–50) relative to the baseline (1961–90) | Literacy (%) |
| Area prone to flood incidence as % geographical area | Change in maximum temperature (°C) during mid-century (2021–50) relative to the baseline (1961–90) | Gender gap defined as the difference between total literacy and female literacy |
| Drought proneness computed by combining the probability of occurrence of severe and moderate droughts | Change in minimum temperature (°C) during mid-century (2021–50) relative to the baseline (1961–90) | Access to markets defined as number of agricultural markets per 1 lakh holdings |
| Available water holding capacity of the soil defined as the amount of water that the soil can hold (mm) | Change in incidence of extremely hot days during March to May when temperature exceeds the normal by 4°C at least during mid-century (2021–50) relative to the baseline (1961–90) | Road connectivity defined as % of villages that have paved roads in relation to total number of villages |
| Stage of groundwater development (Ratio of draft to availability) | Change in incidence of extremely cold days during December to February when temperature falls below the normal by 4°C at least during mid-century (2021–50) relative to the baseline (1961–90) | Rural electrification defined as number of villages with electricity supply in relation to total number of villages (%) |
| Rural population density defined as number of rural people per square km of geographical area | Change in frequency of occurrence of frost days (during Dec-Feb) during mid-century (2021–50) relative to the baseline (1961–90) | Net irrigated area defined as % of net sown area with access to irrigation |
| Area owned by small and marginal farmers in relation to total sown area (%) | Change in drought proneness during mid-century (2021–50) relative to the baseline (1961–90) | Density of livestock defined as number of livestock (small and large ruminants) expressed in terms of adult cattle units per sq. km of geographical area |
| | Change in incidences of dry spells of ≥ 14 days during June to October during mid-century (2021–50) relative to the baseline (1961–90) | Fertilizer consumption (N + P + K) per ha of gross sown area |
| | Extreme rainfall events represented through four different indicators: change (%) in 99 percentile rainfall, change (%) in number of events with > 100 mm rainfall in 3 days, change in mean maximum rainfall in single day as % to annual normal, and change in mean maximum rainfall in 3 consecutive days as % to annual normal during mid-century (2021–50) relative to the baseline (1961–90) | Groundwater availability (ha m/sq. km) |
| | | Share of agriculture in district domestic product defined as % of district domestic product contributed by agriculture |

Source: Rao et al. [12]

This was followed by computing the weighted mean of assigned indicators to construct indices for sensitivity, exposure, and adaptive capacity. Lastly, the vulnerability index was computed by taking weighted average of the three indices—with weights of 25, 40 and 35 to exposure, sensitivity and adaptive capacity respectively [12, 19]. All census districts were categorized into five equal quintiles where the districts with top 20% vulnerability score were considered very highly vulnerable and those in the bottom 20% were considered as very low vulnerable. More information on the various definitions, formulas, and weights used to compute component-wise and vulnerability index can be found in detail in the study report [19].

**National Family Health Survey-4 (NFHS-4).** The Indian equivalent of the Demographic and Health Survey (DHS)—NFHS is conducted at regular intervals to generate information on various fertility, mortality, child health, and nutrition indicators at the district, state, and national levels. The fourth round of NFHS was conducted in 2015–16 and 699,686 women aged 15–49 years old were interviewed from 601,509 households across all states and union territories (UTs) of India. Data on stunting, wasting and underweight for 243,213 children and anaemia for 216,049 children born to ever-married women in the last five years preceding the survey was available. The women were recruited through a stratified two-stage sampling process. In the first stage, primary sampling units (PSUs) were selected systematically using a probability proportional to size approach, and a fixed number of households and eligible women were selected within the PSUs. In rural areas, a village was considered as the PSU, whereas in urban areas it was a census enumeration block. More information on the sampling procedure along with the distribution of socio-demographic, household-level and individual-level characteristics at the state as well as district level can be found in the NFHS-4 Reports [21].

**Matching vulnerability index data with NFHS-4.** While the vulnerability index was computed for 572 districts as per Census 2001, NFHS-4 provided information on the nutritional status of children under five years of age for all 640 districts as listed in Census 2011. Therefore, to conduct the analysis, a district-level mapping exercise was carried out. A list of 572 districts, for which vulnerability index data was computed, was first matched with NFHS districts based on the district/town names. Districts that were common across both data were assigned the corresponding overall vulnerability, sensitivity, exposure, and adaptive capacity indices. For newly formed districts that were available in NFHS-4 data but not in the vulnerability data, indices corresponding to their origin district were assigned. For example, Anjaw district of Arunachal Pradesh was assigned the indices corresponding to its origin district Lohit as available in the vulnerability data. In four instances where new districts were carved out from more than one Census 2001 district, all four indices for newly formed districts were computed by calculating the median of origin district indices. All 16 metropolitan cities/ UTs for which vulnerability index was not available were excluded from the analysis. Following the assumption that these 16 districts were not considered as they are mostly urban and may not have relevant indicators required for constructing the index, 10 more districts were dropped from the remaining UTs. This resulted in observations from 614 districts of all states sans UTs. After these matching, the district level vulnerability map was recreated for the 614 districts (S1 Fig) and compared with the map based on 572 districts created originally by Rao et al. [19] and found no difference in district categorization.

## Ethics statement

The authors did not collect any primary data for this study. Further, the climate change vulnerability index did not include any data collected from human participants. The nodal agency for collecting NFHS-4 data was International Institute for Population Sciences (IIPS), Mumbai. The protocol for NFHS-4 data collection was approved by institutional review boards of IIPS and ORC Macro.

## Measures

**Nutritional status outcomes.** Among all living children under the age of 5 years, nutritional status outcomes considered for this study were stunting, severe stunting, wasting, severe wasting, underweight, severe underweight, anaemia, multiple malnutrition, and all forms of malnutrition. DHS definitions per the World Health Organization's (WHO) child growth standard were used to compute measures on children's nutritional status. Any child whose height-

for-age z score was below minus 2 (-2.0) SD of the mean value was defined as stunted, whereas a child with height-for-age z score below -3.0 SD of the mean was defined as severely stunted. A child was defined as wasted if his/her weight-for-height z score was below -2.0 SD of the mean value. Severely wasted children had a weight-for-height z score below -3.0 SD of the mean. Any child whose weight-for-age z score was below -2.0 SD of the mean value was defined as underweight, whereas a child with a weight-for-age z score below -3.0 SD of the mean was defined as severely underweight. Children aged 6–59 months who stayed in the household the night before the interview with haemoglobin count lower than 11.0 grams per decilitre (g/dl) were defined as anaemic. All living children under the age of 5 years were defined to have multiple malnutrition if out of the four considered nutritional outcomes—stunting, wasting, underweight, and anaemia—they had at least two. If a child was stunted, wasted, underweight as well as anaemic s/he was defined to have all forms of malnutrition. The socio-economic and demographic characteristics that were used as covariates in multivariable analyses are religion, caste, wealth index, place of residence of the household, number of household members, age of the child, sex of the child, mother's education, and birth order. These variables were recoded from the original questions to make them suitable for the present analysis.

## Statistical analyses

Bivariate and multivariable analyses were conducted to examine the association of degree of vulnerability with the nutritional status of children. Spatial analysis was also conducted to understand the macro-level association and spill-over effect a district's climate vulnerability can have on child malnutrition. The analysis was started by conducting bivariate analysis between the degree of vulnerability and nutritional status of children. To answer the first research question, mixed-effect multilevel models were fitted to examine the strength of association between vulnerability and child nutrition. In the mixed-effect model, births were nested within primary sampling units (as defined in NFHS-4 data), which were nested within a district and controlled for socio-demographic, household, and maternal characteristics.

Spatial analysis was conducted at the district-level where child malnutrition indicators were transformed into proportions. First, spatial autocorrelation was computed using Moran's I and Geary's C to understand the extent of spatial clustering in child malnutrition and climate vulnerability. Both these indices provide an idea on the extent to which a spatial regression is suitable. The Moran's I value ranges from -1 to +1 where a positive value indicates positive spatial autocorrelation, and a negative value indicates the negative autocorrelation. Higher the absolute Moran's I value, stronger is the spatial autocorrelation and vice-versa [22]. The Geary's C ranges from 0 to 2; where 1 is no spatial autocorrelation, values near 0 are positively spatially correlated and those closer to 2 are highly negatively autocorrelated. Additionally, hotspots and coldspots were identified using bivariate Local Indicators of Spatial Association (LISA) (Research question # 3). The bivariate LISA generates a choropleth map highlighting the districts with a significant local Moran statistic and classifies them into high-high and low-low spatial clusters, and high-low and low-high spatial outliers. The high-high pairing suggests clustering of values, whereas high-low and low-high locations indicate spatial outliers.

Subsequently, mixed spatial autoregressive error models were fitted for each of the nutrition outcome indicators independently that considered both spatial lag and spatial error. In these spatial regression models, the degree of vulnerability was considered as the key predictor and shares of poor population (head count ratio [23]), proportion of population who belong to rural areas, general caste and Hindu religion were included as covariates. Given that coefficients from a spatial autoregression should not be directly interpreted [24, 25], calculations were within the district (direct) and spill-over (indirect) based on the model coefficients to

answer the second research question. Stata module *spregress* followed by *estat impact* was used to derive these estimates. In addition to the spatial and multivariable analyses, districts burdened with vulnerability and malnutrition were also identified by filtering out districts categorized as having high/very high vulnerability and listing out those districts with child malnutrition levels higher than country average (Research question # 3). Except for the Bivariate LISA, the rest of the analyses were performed using STATA 16.1 (StataCorp., TX, USA). The maps from Bivariate LISA were generated using GeoDa.

## Results

In the study sample, about one-fifth (21%) of children were found to be wasted, two-fifths were stunted (39%) and underweight (36%), and three-fifths had anaemia (59%) (Fig 1). Nearly half of the children (48%) had multiple malnutrition and one in twenty (5%) had all the form of malnutrition.

### Q1. Is there an association between the degree of vulnerability in agriculture to climate change and child nutrition at micro-level?

The degree of vulnerability was positively associated with malnutrition among children (Table 2). For example, children residing in districts with very high degree of vulnerability were more like to have stunting (41% vs 31%, Adjusted Odds Ratio [AOR]: 1.32, 95% CI: 1.21–1.44), wasting (24% vs 19%, AOR: 1.42, 95% CI: 1.27–1.60), underweight (39% vs 30%, AOR: 1.45, 95% CI: 1.30–1.61) and anaemia (63% vs 52%, AOR: 1.75, 95% CI: 1.47–2.08) than those living in districts considered to have very low degree of vulnerability. The magnitude of difference between very high and very low degree of vulnerability was observed to be higher for children severely stunted, severely wasted and severely underweight.

### Q2. Is there any spatial effect of climate vulnerability on child nutrition?

The spatial autocorrelation assessed using Moran's I and Geary's C suggests that there is clear evidence of geographic clustering in both nutrition indicators and degree of vulnerability (Table 3). The evidence of clustering was found to be strongest for children being underweight,

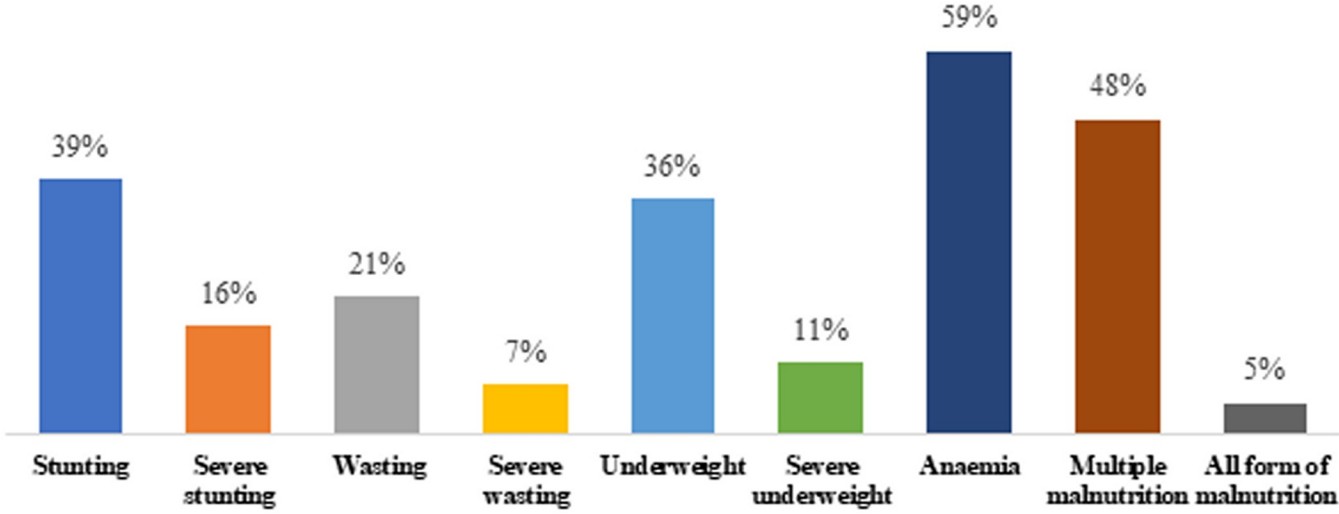

**Fig 1. Prevalence of various nutritional status indicators among children aged five or below.**

**Table 2. Unadjusted percentage and adjusted odds ratio predicting the nutritional status of children by the degree of climate vulnerability, India.**

| | Degree of climate vulnerability | | | | | Intra-class correlation coefficient | |
|---|---|---|---|---|---|---|---|
| | Very low | Low | Moderate | High | Very high | District | District>PSU |
| **Stunting** | | | | | | | |
| % (N = 211,748) | 31.4 | 36.5 | 38.9 | 43.1 | 40.9 | | |
| AOR (95% CI) | Referent | 1.15 (1.05–1.26) | 1.25 (1.15–1.37) | 1.34 (1.23–1.46) | 1.32 (1.21–1.44) | 0.02 | 0.10 |
| **Severe stunting** | | | | | | | |
| % (N = 211,748) | 11.4 | 14.3 | 16.2 | 19.8 | 18.5 | | |
| AOR (95% CI) | Referent | 1.16 (1.04–1.30) | 1.32 (1.18–1.47) | 1.51 (1.36–1.69) | 1.54 (1.38–1.71) | 0.03 | 0.14 |
| **Wasting** | | | | | | | |
| % (N = 211,748) | 19.2 | 20.3 | 20.1 | 21.5 | 23.6 | | |
| AOR (95% CI) | Referent | 1.02 (0.90–1.15) | 1.12 (0.99–1.26) | 1.25 (1.11–1.41) | 1.42 (1.27–1.60) | 0.04 | 0.15 |
| **Severe wasting** | | | | | | | |
| % (N = 211,748) | 6.2 | 7.0 | 7.1 | 7.6 | 8.8 | | |
| AOR (95% CI) | Referent | 1.06 (0.90–1.26) | 1.22 (1.04–1.43) | 1.35 (1.15–1.58) | 1.61 (1.38–1.88) | 0.05 | 0.26 |
| **Underweight** | | | | | | | |
| % (N = 211,748) | 30.0 | 33.7 | 35.5 | 40.0 | 38.6 | | |
| AOR (95% CI) | Referent | 1.08 (0.97–1.21) | 1.27 (1.14–1.41) | 1.44 (1.30–1.60) | 1.45 (1.30–1.61) | 0.03 | 0.11 |
| **Severe underweight** | | | | | | | |
| % (N = 211,748) | 8.1 | 9.9 | 10.6 | 13.0 | 12.9 | | |
| AOR (95% CI) | Referent | 1.13 (1.00–1.29) | 1.26 (1.11–1.43) | 1.49 (1.32–1.69) | 1.61 (1.43–1.82) | 0.03 | 0.16 |
| **Anaemia** | | | | | | | |
| % (N = 197,186) | 52.3 | 54.4 | 58.8 | 61.0 | 62.6 | | |
| AOR (95% CI) | Referent | 1.02 (0.85–1.22) | 1.43 (1.20–1.71) | 1.59 (1.33–1.89) | 1.75 (1.47–2.08) | 0.10 | 0.21 |
| **Multiple malnutrition** | | | | | | | |
| % (N = 188,930) | 38.9 | 44.1 | 47.2 | 52.2 | 51.6 | | |
| AOR (95% CI) | Referent | 1.12 (0.99–1.26) | 1.38 (1.23–1.55) | 1.58 (1.41–1.77) | 1.65 (1.47–1.85) | 0.04 | 0.13 |

Note: Mixed effect multilevel model adjusted for religion, caste, wealth index, place of residence, number of household members, age of the child, sex of the child, mother's education, birth order

followed by stunting and anaemia. The spatial autoregressive model suggests that malnutrition among children is likely to be more in districts that are very highly vulnerable to climate compared to those that have a very low degree of vulnerability (Table 4). For example, stunting is likely to be 3% more in very highly vulnerable districts than those with very low vulnerability. Similarly, compared to districts categorised as very low in terms of vulnerability, children from the very high category are 4% more likely to have wasting and underweight, and 6% more likely to have anaemia. Similar within district effects were noted for those districts with high vulnerability. The study also examined if the district's vulnerability has a spill-over across districts. Districts categorized as very high vulnerability were also found to be more likely to have a spill-over effect across the neighbouring districts. For example, districts with very high vulnerability are likely to have a spillover effect of stunting by 0.24 percentage point compared to very low vulnerability district.

## Q3. Which are the geographical hotspots of the degree of vulnerability in agriculture to climate change and child malnutrition?

The Bivariate LISA maps (Fig 2) show the hotspots and coldspots in the spatial relationship between the degree of vulnerability and child malnutrition indicators. The number of high-

**Table 3. Moran's I and Geary's C values assessing degree of autocorrelation in degree of vulnerability and nutrition status of children.**

| | Moran's I | | |
|---|---|---|---|
| **Indicators** | I | Z-Value | P-Value |
| Very high degree of vulnerability | 0.459 | 18.8 | <0.001 |
| Stunting | 0.643 | 26.3 | <0.001 |
| Severe stunting | 0.571 | 23.3 | <0.001 |
| Wasting | 0.500 | 20.5 | <0.001 |
| Severe wasting | 0.273 | 11.2 | <0.001 |
| Underweight | 0.730 | 29.8 | <0.001 |
| Severe underweight | 0.627 | 25.6 | <0.001 |
| Anaemia | 0.617 | 25.2 | <0.001 |
| | **Geary's C** | | |
| | C | Z-Value | P-Value |
| Very high degree of vulnerability | 0.591 | -13.2 | <0.001 |
| Stunting | 0.315 | -24.1 | <0.001 |
| Severe stunting | 0.397 | -19.9 | <0.001 |
| Wasting | 0.495 | -16.6 | <0.001 |
| Severe wasting | 0.708 | -7.9 | <0.001 |
| Underweight | 0.240 | -27.0 | <0.001 |
| Severe underweight | 0.358 | -21.2 | <0.001 |
| Anaemia | 0.367 | -21.0 | <0.001 |

high clusters varied across child nutrition indicators: 92 for underweight, 79 for stunting, 75 for wasting and 65 for anaemia. Similarly, the number of low-low clusters were highest for underweight (113) and least for wasting (82). The LISA maps suggest that hotspots of child malnutrition and degree of vulnerability are mostly clustered around western-central part of India though there were some hotspots for stunting in the eastern part of the country as well. Further drill-down of the district-level data found a total of 69 districts that had high levels of stunting, wasting, underweight and anaemia together with high/very high level of vulnerability (S1 Table). These districts belonged to the states of Bihar, Chhattisgarh, Gujarat, Haryana, Jharkhand, Karnataka, Madhya Pradesh, Maharashtra, Rajasthan, and Uttar Pradesh (Table 5).

## Discussion

Climate scientists have predicted that climate change is going to have a significant impact on agriculture which will ultimately affect the quality and quantity of food production [1, 3]. This study examined how the vulnerability of a district to climate can affect child nutrition. The study found that districts highly vulnerable to climate change can have more child malnutrition than districts which are relatively less vulnerable. The mixed-effect analysis found that the odds of a child suffering from stunting increased by 32%, wasting by 42%, underweight by 45% and anaemia by 63% if the child belonged to a district categorised as very highly vulnerable when compared to those categorised as very low. The magnitude of effects was stronger when examined for severe- stunting, wasting and underweight. The macro-level spatial analysis demonstrated that rates of child malnutrition were higher by 3–5% for very highly vulnerable districts than very low vulnerable ones. The study also investigated if the effect of high/very high vulnerability on child nutrition transferred to neighbouring districts and found significant evidence of spill-over for stunting but not for wasting, underweight and anaemia. Lastly, the study used bivariate spatial maps and macro-level data to identify the clusters where child malnutrition

**Table 4. Expected percentage gain/reduction in nutritional status of children within and across districts by degree of vulnerability estimated using spatial autoregressive model.**

| | Degree of climate vulnerability | | | | |
|---|---|---|---|---|---|
| | Very low | Low | Moderate | High | Very high |
| **Stunting** | | | | | |
| Within-district direct effect | Referent | 1.32 (-0.26–2.91) | **2.25 (0.54–3.96)** | **3.08 (1.23–4.92)** | **3.50 (1.49–5.50)** |
| Across district spill-over effect | Referent | 0.09 (-0.04–0.22) | 0.16 (-0.01–0.32) | **0.21 (0.01–0.41)** | **0.24 (0.03–0.46)** |
| Auto-correlation | 0.73, P<0.001 | | | | |
| **Severe stunting** | | | | | |
| Within-district direct effect | Referent | 0.48 (-0.67–1.64) | **1.6 (0.36–2.84)** | **3.00 (1.67–4.34)** | **3.37 (1.92–4.81)** |
| Across district spill-over effect | Referent | 0.05 (-0.07–0.17) | 0.16 (-0.02–0.34) | **0.30 (0.01–0.58)** | **0.33 (0.03–0.64)** |
| Auto-correlation | 0.65, P<0.001 | | | | |
| **Wasting** | | | | | |
| Within-district direct effect | Referent | 0.40 (-1.12–1.91) | 0.53 (-1.09–2.15) | **2.37 (0.64–4.10)** | **3.62 (1.76–5.48)** |
| Across district spill-over effect | Referent | 0.04 (-0.13–0.21) | 0.06 (-0.12–0.24) | 0.26 (-0.02–0.53) | **0.39 (0.03–0.75)** |
| Auto-correlation | 0.56, P<0.001 | | | | |
| **Severe wasting** | | | | | |
| Within-district direct effect | Referent | 0.03 (-0.93–0.99) | 0.67 (-0.34–1.67) | **1.24 (0.17–2.31)** | **2.42 (1.30–3.54)** |
| Across district spill-over effect | Referent | 0.001 (-0.13–0.13) | 0.09 (-0.07–0.25) | 0.17 (-0.05–0.39) | 0.33 (-0.04–0.69) |
| Auto-correlation | 0.34, P<0.001 | | | | |
| **Underweight** | | | | | |
| Within-district direct effect | Referent | 0.69 (-0.95–2.32) | **2.13 (0.36–3.90)** | **4.09 (2.18–6.00)** | **3.83 (1.74–5.92)** |
| Across district spill-over effect | Referent | 0.07 (-0.11–0.26) | **0.23 (0.002–0.46)** | **0.44 (0.13–0.76)** | **0.42 (0.11–0.72)** |
| Auto-correlation | 0.75, P<0.001 | | | | |
| **Severe underweight** | | | | | |
| Within-district direct effect | Referent | 0.29 (-0.61–1.19) | **0.99 (0.03–1.96)** | **2.02 (0.99–3.05)** | **2.91 (1.81–4.02)** |
| Across district spill-over effect | Referent | 0.07 (-0.14–0.28) | 0.23 (-0.01–0.47) | **0.47 (0.16–0.78)** | **0.68 (0.29–1.06)** |
| Auto-correlation | 0.57, P<0.001 | | | | |
| **Anaemia** | | | | | |
| Within-district direct effect | Referent | -0.1 (-2.78–2.57) | 2.57 (-0.37–5.5) | **3.61 (0.41–6.82)** | **4.91 (1.35–8.47)** |
| Across district spill-over effect | Referent | -0.01 (-0.15–0.14) | 0.14 (-0.09–0.36) | 0.19 (-0.09–0.48) | 0.26 (-0.09–0.62) |
| Auto-correlation | 0.83, P<0.001 | | | | |

Note: Spatial autoregressive models were adjusted for proportion poor, living in rural areas, and belonging to general caste and Hindu religion.

and vulnerability were high. Further, the study identified 69 districts that were battling the double burden of high/very high climate vulnerability as well as child malnutrition.

India being the second largest populous country with a heavy dependency on agriculture, high vulnerability of certain regions to climate change can be cause of concern to agriculturalists and policymakers [26]. Though the country has seen significant economic development in the last couple of decades, similar progress has not been made in addressing child malnutrition [27]. Child malnutrition is prevalent across states whether they are at the forefront of economic development (e.g. Gujarat) or lagging (e.g. Bihar, and Uttar Pradesh) [28]. While the study provides indisputable evidence on effect agriculture's vulnerability to climate change, this effect may be further explained by inadequate health infrastructure and poverty. A closer look at the 69 districts facing the double burden of climate vulnerability and child malnutrition suggests that most of these districts and states are characterized by poor health infrastructure in rural areas, low literacy, rudimentary sanitation, and poverty. A study by Khan and Mohanty has highlighted how poverty has a significant impact on child malnutrition in India

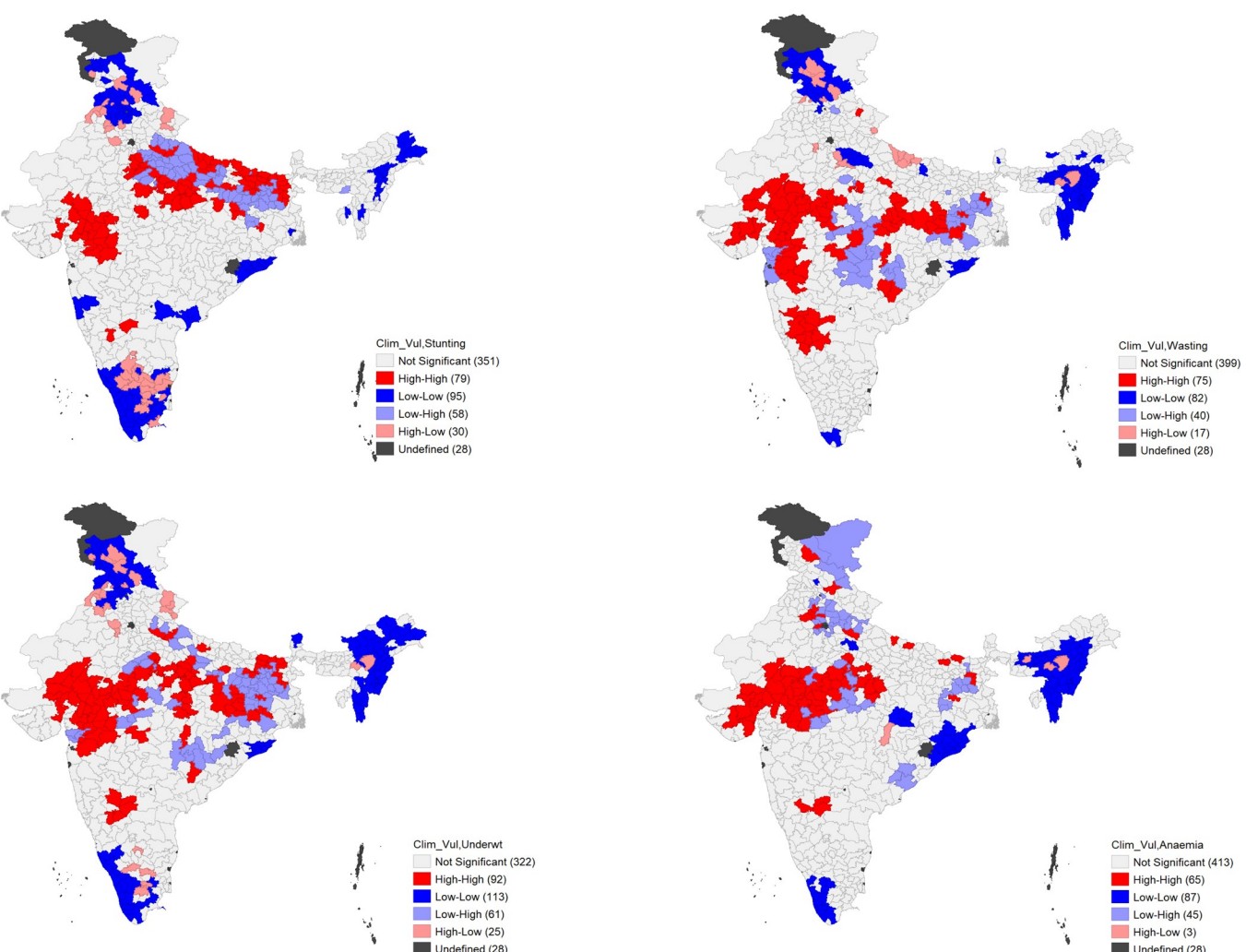

**Fig 2. Bivariate LISA-based maps highlighting hotspots and coldspots in spatial correlation between the degree of vulnerability and child malnutrition.**

[28]. Consistent with earlier studies, the hotspots of child malnutrition and degree of vulnerability are concentrated in the areas where hotspots of poverty and child malnutrition have been identified. This suggests a close relationship between the degree of vulnerability and poverty level which should be explored further in future research. The clustering of vulnerability levels and child malnutrition indicates the extent to which climate change can affect the food production system and ultimately the nutrition of children in the short run and adults in the long run. However, the early evidence from this study provided an opportunity to governments and programmers to develop sustainable solutions towards mitigating the effects that climate change will have on agriculture and human health.

Of the notable findings in this study is the estimation of within-district and spill-over effect of climate vulnerability on child malnutrition. Among all the malnutrition indicators, the effect of vulnerability was most on anaemia (5% [within-district + spill-over]), followed by underweight, stunting and wasting (4%). Notably, the malnutrition indicators had higher spatial autocorrelation suggesting geographical clustering. Within-district effects of climate vulnerability, particularly for high and very highly vulnerable districts were substantially significant. This indicated that there will be a significant effect of climate vulnerability on child malnutrition

**Table 5. Number of districts by state which have child malnutrition levels higher than India average in districts categorized as "high" or "very high" in degree of vulnerability.**

| Stunting | Wasting | Underweight | Anaemia |
|---|---|---|---|
| Andhra Pradesh (2) | Bihar (7) | Andhra Pradesh (2) | Andhra Pradesh (1) |
| Assam (1) | Chhattisgarh (5) | Bihar (19) | Bihar (18) |
| Bihar (19) | Gujarat (19) | Chhattisgarh (5) | Chhattisgarh (1) |
| Chhattisgarh (4) | Haryana (5) | Gujarat (14) | Gujarat (13) |
| Gujarat (12) | Jharkhand (12) | Haryana (2) | Haryana (9) |
| Haryana (1) | Karnataka (16) | Jharkhand (12) | Himachal Pradesh (2) |
| Jharkhand (12) | Madhya Pradesh (30) | Karnataka (13) | Jammu & Kashmir (1) |
| Karnataka (9) | Maharashtra (14) | Madhya Pradesh (32) | Jharkhand (11) |
| Madhya Pradesh (23) | Odisha (1) | Maharashtra (12) | Karnataka (15) |
| Maharashtra (9) | Punjab (2) | Odisha (1) | Madhya Pradesh (33) |
| Rajasthan (15) | Rajasthan (20) | Rajasthan (19) | Maharashtra (4) |
| Uttar Pradesh (27) | Tamil Nadu (6) | Uttar Pradesh (22) | Odisha (1) |
|  | Uttar Pradesh (11) | Uttarakhand (1) | Punjab (2) |
|  | Uttarakhand (2) | West Bengal (1) | Rajasthan (16) |
|  | West Bengal (1) |  | Uttar Pradesh (22) |
|  |  |  | Uttarakhand (1) |

among districts categorized as very high/high, irrespective of the neighbouring districts' vulnerability level. The spill-over effect of vulnerability was significant for all malnutrition indicators except for anaemia. This again highlighted that the effect of vulnerability is not limited by the geographical boundaries rather the effect can extend to neighbouring districts as well. Interestingly, the spill-over was not present when severe malnutrition was examined.

The findings of the study should be interpreted in the light of following limitations. First, union territories and completely urban districts were excluded from the analysis as the vulnerability index values were not available for those areas. Second, the original index was based on 572 districts which re-mapped into 614 districts, as a result some of the district's vulnerability ranking may have been wrongly assigned. However, it is assumed that such misplacing would be very minimal and not likely to change the results presented in the study. To ensure that mapping of degree of vulnerability is robust, the vulnerability maps provided by Rao et al. [19] for 572 districts were matched with the one generated for 614 districts. Third, the study did not examine the dietary intake pattern (both quantity and quality) of children and their families which is likely to have an influence on their nutritional status. Future research should collect dietary intake data and examine if vulnerability to climate change has an influence on dietary intake and whether the pattern of consumption play a role in determining the relationship between vulnerability and nutritional status. Lastly, obtaining data on all the variables/indicators for a uniform reference period at the district-level is extremely difficult. While vulnerability index computation used the most recent data available for each unit of analysis, for missing data statistical methods such as using nearest neighbourhood value, average value of respective state, simulation and extrapolation methods were used to derive the indicators at the district level for computing vulnerability index. While not a limitation to this study, it is also to be noted that the vulnerability index created were assigned unequal weights to the three dimensions of adaptive capacity, exposure, and sensitivity. Though unequal weight assignment is well justified by the authors [12, 19], it would have been worth exploring how the vulnerability index would look if equal weights were assumed and how that, in turn, would affect the evidence generated by the study.

The study has important implications for both research and policy to address climate vulnerability and child malnutrition. Existing and future programs in India, specifically those

focussing on nutrition and agriculture, should consider the vulnerability of agriculture to climate change in developing their strategies. For areas where agriculture is vulnerable to climate change, there should be increasing efforts to grow staple crops that can sustain in given climatic conditions as well as meet the nutritional requirements of the population. Given that the current research identifies such geographic cluster, it would be important to develop cluster-specific agricultural plans based on the nutritional requirements of the area. While this study identified clusters of geographies where vulnerability and malnutrition exist, it would be important to further drill down and identify the sub-clusters (sub-district or *panchayat*) within those areas where the problem lies. This will help more specific targeted programming for agriculture and providing nutrition supplements to children. While this study identified the effect of vulnerability to climate change on child malnutrition, future research should explore whether the climate vulnerability has an impact on adults' nutritional status and other co-morbidities emerging from malnutrition. In conclusion, this is the first study to examine the relationship between the degree of vulnerability in agriculture to climate change and child malnutrition. The study found strong evidence at both micro and macro levels on how the vulnerability of agriculture to climate change can result in child malnutrition. The clustering of vulnerability and child malnutrition at few select states and districts that are historically known for multiple deprivations further highlights the need to have a holistic approach to bring change in the lives of people living in those geographical areas. Finally, this effect of climate vulnerability is not limited to that district, but it spills to the adjoining areas as well.

## Supporting information

**S1 Fig. Degree of vulnerability of agriculture to climate change at district level.**
(TIF)

**S1 Table. Districts categorized as "high" or "very high" in degree of vulnerability and having child malnutrition levels higher than India average.**
(DOCX)

## Author Contributions

**Conceptualization:** Bidhubhusan Mahapatra, Chitiprolu Anantha Rama Rao.

**Data curation:** Monika Walia, Bellapukonda Murali Krishna Raju.

**Formal analysis:** Bidhubhusan Mahapatra, Monika Walia, Bellapukonda Murali Krishna Raju.

**Investigation:** Chitiprolu Anantha Rama Rao.

**Methodology:** Bidhubhusan Mahapatra, Niranjan Saggurti.

**Software:** Bidhubhusan Mahapatra.

**Supervision:** Niranjan Saggurti.

**Validation:** Chitiprolu Anantha Rama Rao.

**Visualization:** Bidhubhusan Mahapatra, Chitiprolu Anantha Rama Rao.

**Writing – original draft:** Bidhubhusan Mahapatra.

**Writing – review & editing:** Monika Walia, Chitiprolu Anantha Rama Rao, Niranjan Saggurti.

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
