## [Decision Letter · Decision Letter 0]

6 Apr 2021

PONE-D-21-00139

Vulnerability of agriculture to climate change increases the risk of child malnutrition: Evidence from a large-scale observational study in India

PLOS ONE

Dear Dr. Mahapatra,

Thank you for submitting your manuscript to PLOS ONE. After careful consideration, we feel that it has merit but does not fully meet PLOS ONE’s publication criteria as it currently stands. Therefore, we invite you to submit a revised version of the manuscript that addresses the points raised during the review process.

Considering the reviewer's suggestion, I am going with a decision of 'Revise and Resubmit'.

In particular, reviewers believe that although the paper is interesting, the validity of the vulnerability index needs to be established.

I have an additional comment for this paper. Considering that weather stations to measure different climatic indicators are not present in the every district, rather they are very few in rural areas. Thus, researchers depend on simulation and extrapolation methods to derive the indicators at the district level. In this process, there is a chance of errors that need to be examined and reported under the limitation section. I request you to add a limitation section for the paper and validate climatic data.

We look forward to receiving your revised manuscript.

Kind regards,

Srinivas Goli, Ph.D.

Academic Editor

PLOS ONE

Journal Requirements:

Please add details of how others can access the climate dataset in the Data availability statement.

We note that Figure S1 and Figure 2 in your submission contain map images which may be copyrighted. All PLOS content is published under the Creative Commons Attribution License (CC BY 4.0), which means that the manuscript, images, and Supporting Information files will be freely available online, and any third party is permitted to access, download, copy, distribute, and use these materials in any way, even commercially, with proper attribution. For these reasons, we cannot publish previously copyrighted maps or satellite images created using proprietary data, such as Google software (Google Maps, Street View, and Earth). For more information, see our copyright guidelines: http://journals.plos.org/plosone/s/licenses-and-copyright.

3a, You may seek permission from the original copyright holder of Figure S1 and Figure 2 to publish the content specifically under the CC BY 4.0 license. 

3b, If you are unable to obtain permission from the original copyright holder to publish these figures under the CC BY 4.0 license or if the copyright holder’s requirements are incompatible with the CC BY 4.0 license, please either i) remove the figure or ii) supply a replacement figure that complies with the CC BY 4.0 license. Please check copyright information on all replacement figures and update the figure caption with source information. If applicable, please specify in the figure caption text when a figure is similar but not identical to the original image and is therefore for illustrative purposes only.

Additional Editor Comments:

Considering the reviewer's suggestion, I am going with a decision of 'Revise and Resubmit'.

In particular, reviewers believe that although the paper is interesting, the validity of the vulnerability index needs to be established.

I have an additional comment for this paper. Considering that weather stations to measure different climatic indicators are not present in the every district, rather they are very few in rural areas. Thus, researchers depend on simulation and extrapolation methods to derive the indicators at the district level. In this process, there is a chance of errors that need to be examined and reported under the limitation section. I request you to add a limitation section for the paper and validate climatic data.

Reviewers' comments:

Reviewer's Responses to Questions

**Comments to the Author**

1. Is the manuscript technically sound, and do the data support the conclusions?

Reviewer #1: Yes

Reviewer #2: Yes

2. Has the statistical analysis been performed appropriately and rigorously? 

Reviewer #1: N/A

Reviewer #2: Yes

3. Have the authors made all data underlying the findings in their manuscript fully available?

Reviewer #1: Yes

Reviewer #2: Yes

4. Is the manuscript presented in an intelligible fashion and written in standard English?

Reviewer #1: Yes

Reviewer #2: Yes

5. Review Comments to the Author

Reviewer #1: Generally, the topic of this manuscript is of high priority because it discusses the consequences of global warming threats on child malnutrition in India. Authors used an index on vulnerability of agriculture to climate change and linked it to child malnutrition indicators (stunting, wasting, underweight and anaemia) from the National Family Health Survey 4 (2015-16) and concluded that the global warming threat has negative impacts on child nutrition. However, I think authors should consider two main points before accepting this manuscript:

L 32-33....the aim of this study that was written in the introduction is not acceptable. , the global warming treat negatively affects human nutrition all over the world not only in India. This is not the first study to examine such a relationship; however, the measuring tools and the data analyses style might provide an advantage for this research over many others. I think authors should re-write this aim

The second point is related to the measuring tools that are used in this study. I think they should be rewritten in details.

For these two reasons, I recommend accepting this manuscript; after considering minor revision

Reviewer #2: This is an interesting manuscript and possibly first attempt to establish association of under-nutrition and vulnerability of agriculture climate change. While there are interesting finding, the manuscript need improvement in introduction, analyses and presentation. The following are major comments on the paper

1. Rewrite introduction. Brin first sentence (line no 302-04) to the begining of introduction. Line 53, introduction, avoid giving magnitude by 2080- which is very long. Put till 2050. line 60, avoid 10-60% give number

2. Para on India in intro and para on India in discussion can be combined and placed after literature review in intro

3.Review, 13-17, give finding and not listing studies

4.Data source: The climate vulnerability index is novelty of the paper. Describe adequately as many readers are not aware of it. Give a two way graph of vulnerability index with mean years of schooling (nfhs 4) or with Dist Devl Index by Mohanty et al to check its association with devl.

5. Describe detail of variables in table S1

6. Figure S1 and a map on stunting at dt level map be begining . Also give correlation coeffcient of these two

7. Table 1: I suppose t1 has been derived after merging dist index value in data file. In such cases the index is a constant across district. I suggest table 1 may be presented as dt level analyses and dt is the unit of analyses. It may be appropriate to do so

8. Mixed effect model, can you show the VPC?

9.T2 is univariate moran I?

10. How classification of very low----- very high of vulnerability index made

11. Authors may consider reduction in length of title

6. PLOS authors have the option to publish the peer review history of their article (what does this mean?). If published, this will include your full peer review and any attached files.

Reviewer #1: No

Reviewer #2: **Yes: **Sanjay K Mohanty

---

## [Author Response · Author response to Decision Letter 0]

11 May 2021

Editor’s comments

Considering that weather stations to measure different climatic indicators are not present in the every district, rather they are very few in rural areas. Thus, researchers depend on simulation and extrapolation methods to derive the indicators at the district level. In this process, there is a chance of errors that need to be examined and reported under the limitation section. I request you to add a limitation section for the paper and validate climatic data.

Response: Many thanks for bringing this to our attention. We have included the non-availability of climatic indicators in every district as a limitation of the vulnerability index under limitation section.

Reviewers' comments:

Reviewer #1: 

1. L 32-33....the aim of this study that was written in the introduction is not acceptable, the global warming treat negatively affects human nutrition all over the world not only in India. This is not the first study to examine such a relationship; however, the measuring tools and the data analyses style might provide an advantage for this research over many others. I think authors should re-write this aim.

Response: We agree that our study is not the first one in India to study the negative effects of global warming, however, as also highlighted by Reviewer #2, it is the first of its kind wherein relationship between vulnerability of agriculture to climate change and child nutrition in India using vulnerability index has been studied. We have changed the text in introduction section of the abstract to as per the suggestion.

2. The second point is related to the measuring tools that are used in this study. I think they should be rewritten in details.

Response: As suggested, we have provided more details about the measurement process and indicators. Accordingly, we have moved the supplementary table on indicators (Table S1) as a main table (Table 1) in the revised paper.

Reviewer #2: 

1. Rewrite introduction. Brin first sentence (line no 302-04) to the begining of introduction. \\

Line 53, introduction, avoid giving magnitude by 2080- which is very long. Put till 2050. line 60, avoid 10-60% give number

Response: Thank you for your suggestion on rewriting introduction. As suggested, we have moved first line from discussion section to the beginning of introduction. Regarding, providing the estimate till year 2050 instead of 2080, as climatic changes take longer to reflect and transform, the models and predictions studying the impact over next 50 years or more tend to be more reliable. Also, since 2080 is only 59 years away, we feel the estimates for 2080 aren’t very far off in future.

2. Para on India in intro and para on India in discussion can be combined and placed after literature review in intro.

Response: After reviewing the suggestion by the reviewer, we feel the para on India is better suited in discussion. Therefore, we have retained the paragraph on India in the discussion as it summarizes the situation in national context well and establishes a linkage between vulnerability of agriculture to climate change and child malnutrition. 

3.Review, 13-17, give finding and not listing studies

Response: Thank you for the suggestion. We have added summary of findings from the studies cited in reference 13-17 in the introduction.

4.Data source: The climate vulnerability index is novelty of the paper. Describe adequately as many readers are not aware of it. 

Response: Thank you for your suggestion. While reviewer thinks that vulnerability index is the novelty of the paper, we have a different opinion as the vulnerability index is already published in a separate form and this paper only extends the analysis by studying the impact of vulnerability of agriculture to climate change. Nevertheless, as suggested, we have revised the methods section to provide more details about the climate vulnerability index. 

5. Give a two way graph of vulnerability index with mean years of schooling (nfhs 4) or with Dist Devl Index by Mohanty et al to check its association with devl.

Response: We feel the linkages between climate vulnerability and development do not align with study’s objective. Also, the rationale of two-way graph with mean years of schooling is not clear. Therefore, we have not added the suggested graph.

6. Describe detail of variables in table S1 

Response: Thank you for the suggestion. We have tried to elaborate the indicator definitions. We have also moved Table S1 to the manuscript as Table 1.

7. Figure S1 and a map on stunting at dt level map be beginning. Also give correlation coeffcient of these two

Response: We appreciate the suggestion on including a supplementary map on prevalence of stunting at the district level. Since the district level information on stunting (per NFHS) and the relevant maps on stunting are easily available elsewhere we have decided not to include it in the paper. Also, we have several other child nutrition indicators in this paper; hence, giving map only on stunting may not add any further value.

8. Table 1: I suppose t1 has been derived after merging dist index value in data file. In such cases the index is a constant across district. I suggest table 1 may be presented as dt level analyses and dt is the unit of analyses. It may be appropriate to do so

Response: Yes, Table 1 (Now Table 2 in revised version) was derived by combining district level index with child level data. Since the outcome in this case is child nutrition, which is available at the child level, it is more appropriate to undertake the analysis at child level so that the socio-demographic differentials related to the outcome can be captured accurately. 

9. Mixed effect model, can you show the VPC?

Response: As suggested, we have included the intra-class correlation coefficient in the revised manuscript.

10.T2 is univariate moran I? 

Response: Yes, it is univariate.

11. How classification of very low----- very high of vulnerability index made

Response: All the Indian districts were categorized into five equal quintiles. The bottom 20% were considered to have very low vulnerability and the top 20% were considered to have very high vulnerability.

12. Authors may consider reduction in length of title

Response: We think the title is appropriate to reflect the essence of the paper. Therefore, we have refrained from changing the title.

---

## [Decision Letter · Decision Letter 1]

10 Jun 2021

Vulnerability of agriculture to climate change increases the risk of child malnutrition: Evidence from a large-scale observational study in India

PONE-D-21-00139R1

Dear Dr. Mahapatra,

We’re pleased to inform you that your manuscript has been judged scientifically suitable for publication and will be formally accepted for publication once it meets all outstanding technical requirements.

Kind regards,

Srinivas Goli, Ph.D.

Academic Editor

PLOS ONE

Additional Editor Comments (optional):

Considering reviewer recommendation and my own reading of the paper, I am recommending this paper for publication.

Reviewers' comments:

Reviewer's Responses to Questions

**Comments to the Author**

1. If the authors have adequately addressed your comments raised in a previous round of review and you feel that this manuscript is now acceptable for publication, you may indicate that here to bypass the “Comments to the Author” section, enter your conflict of interest statement in the “Confidential to Editor” section, and submit your "Accept" recommendation.

Reviewer #2: All comments have been addressed

2. Is the manuscript technically sound, and do the data support the conclusions?

Reviewer #2: Yes

3. Has the statistical analysis been performed appropriately and rigorously? 

Reviewer #2: Yes

4. Have the authors made all data underlying the findings in their manuscript fully available?

Reviewer #2: Yes

5. Is the manuscript presented in an intelligible fashion and written in standard English?

Reviewer #2: Yes

6. Review Comments to the Author

Reviewer #2: Most of my earlier comments were addressed.

If authors can add the theoretical basis of linking climate change with malnutrition, it would be helpful to reader

7. PLOS authors have the option to publish the peer review history of their article (what does this mean?). If published, this will include your full peer review and any attached files.

Reviewer #2: No

---

## [Editor Report · Acceptance letter]

18 Jun 2021

PONE-D-21-00139R1 

Vulnerability of agriculture to climate change increases the risk of child malnutrition: Evidence from a large-scale observational study in India 

Dear Dr. Mahapatra:

I'm pleased to inform you that your manuscript has been deemed suitable for publication in PLOS ONE. Congratulations! Your manuscript is now with our production department. 

Kind regards, 

on behalf of

Dr. Srinivas Goli 

Academic Editor

PLOS ONE